# Does Physical Activity in Natural Outdoor Environments Improve Wellbeing? A Meta-Analysis

**DOI:** 10.3390/sports10070103

**Published:** 2022-06-29

**Authors:** Caitlin Kelley, Diane E. Mack, Philip M. Wilson

**Affiliations:** Behavioural Health Sciences Research Lab, Department of Kinesiology, Faculty of Applied Health Sciences, Brock University, St. Catharines, ON L2S 3A1, Canada; ckelley@hopeworxinc.org (C.K.); pwilson4@brocku.ca (P.M.W.)

**Keywords:** mental health, green exercise, health promotion, nature, restoration, meta-analysis

## Abstract

Organizational initiatives and researchers have argued for the importance of the natural outdoor environment (NOE) for promoting wellbeing. The main aim of this meta-analysis was to synthesize the existing literature to examine the effects of physical activity (PA) in the NOE on wellbeing in adults. The secondary aim was to explore whether wellbeing reported by adults differs as a function of PA context. Electronic databases (PubMed, ProQuest Nursing and Allied Health, PsycINFO, SPORTDiscus and Embase) were searched for English peer-reviewed articles published before January 2019. Inclusion criteria were: (1) healthy adults; (2) PA in the NOE; (3) the measurement of wellbeing; and (4) randomized control trials, quasi-experimental designs, matched group designs. To address the secondary aim, PA in the NOE was compared with that performed indoors. Risk of bias was assessed through the Effective Public Health Practice Project (EHPP) Quality Assessment Tool for Quantitative Studies. Primary studies meeting inclusion criteria for the main (*n**_studies_* = 19) and secondary (*n**_studies_* = 5) aims were analyzed and interpreted. The overall effect size for the main analysis was moderate (*d* = 0.49, *p* < 0.001; *95% CI* = 0.33, 0.66), with the magnitude of effect varying depending on wellbeing dimension. Wellbeing was greater in PA in the NOE subgroup (*d* = 0.53) when compared with the indoor subgroup (*d* = 0.28), albeit not statistically significant (*p* = 0.15). Although physical activity in the NOE was associated with higher wellbeing, there is limited evidence to support that it confers superior benefits to that engaged indoors. Researchers are encouraged to include study designs that measure markers of wellbeing at multiple time points, greater consideration to diverse wellbeing dimensions and justify decisions linked to PA and NOE types.

## 1. Introduction

Wellbeing has become a focal point of research across the past two decades [1,2]. This is hardly surprising given the empirical links between wellbeing and better health outcomes [3,4] combined with the inclusion of wellbeing as a target for public policy [5,6]. Efforts to understand and enhance wellbeing hold intrinsic and instrumental value, leading researchers to advocate for the evaluation of interventions to improve wellbeing [7]. Physical activity (PA) has been proposed as one intervention platform to improve wellbeing [8]. However, current estimates indicate more than 25% of adults worldwide are physically inactive [9]. This physical inactivity crisis highlights the need to consider approaches for intervention design (and implementation) which can support PA behaviour change, resulting in positive effects on wellbeing. Environmental determinants have been identified as plausible approaches to mitigate physical inactivity behaviour [10], with particular emphasis on the appeal of the natural outdoor environment (NOE) for increasing PA behaviour [11,12]. NOE encompass geographical areas characterized by minimal human presence or intervention (e.g., forests), but can include land use areas where human intrusion has occurred (e.g., parks, sports fields, etc.) [13,14]. Support for the positive effects of performing PA in the NOE has been proposed, including boredom relief [15], lower perceived exertion [16], higher self-esteem and improved mood [12,17]. Furthermore, the beneficial effects of PA in the NOE on wellbeing may be obtained following a single bout [18]. Overall, previous reviews of the literature focusing on PA in the NOE provide evidence favoring greater wellbeing from engagement in these contexts [12,16,17,19], although provide equivocal support for PA in the NOE compared with indoor contexts for enhanced wellbeing [16,18]. Careful inspection of the published research focused on PA in the NOE provided the impetus for this study. First, it is apparent that previous reviews of PA in the NOE have relied on keyword search terms which either (a) have no obvious links to wellbeing (e.g., intention) [16,18], and/or (b) lack content relevance to the domain of wellbeing (e.g., attention, concentration) [19]. Consensus has emerged that wellbeing is multi-faceted and combines hedonic (i.e., feeling good, etc.) and eudaimonic dimensions (i.e., functioning well, etc.) [1,2,20]. Keyword search terms aligned with contemporary views of wellbeing may provide a better understanding of the role ascribed to PA in the NOE for enhancing wellbeing. Second, previous reviews focusing on PA in the NOE have approached study inclusion by relying on (a) research from a limited source (e.g., one research lab) [12], (b) specificity to multi-component interventions that combine PA in the NOE with other therapeutic approaches rendering the unique effects of PA in the NOE on wellbeing undetermined [21], or (c) including studies without placing restrictions on sample characteristics (e.g., children with disabilities) [16,19] that may result in greater between-study heterogeneity which imposes limits on the generalizability of study findings [22]. Based on these issues, combined with the importance of understanding possible routes to enhancing wellbeing that may be amenable to intervention, there is considerable scope for additional scrutiny of the research linking PA in the NOE with subsequent variation in wellbeing. The overarching aim of this investigation was to evaluate the link between PA undertaken in the NOE and wellbeing reported by adults. The main aim of this study was to examine the effects of PA in the NOE on adults’ wellbeing. Building on the work of Barton and Pretty [12], Bowler et al. [19], and Thompson Coon et al. [18], it was hypothesized that PA behaviour in the NOE would positively impact wellbeing. The secondary aim of this study was to investigate whether wellbeing reported by adults differed as a function of being physically active in the NOE versus indoors. Extrapolating from Lahart et al. [16], no differences in wellbeing were expected from PA in the NOE compared with indoor contexts.

## 2. Materials and Methods

This study followed guidelines outlined within the Preferred Reporting Items for Systematic Review and Meta-Analysis Protocols (PRISMA-2020; see Appendix A) [23]. The protocol used in this study was registered in the online database of the International Prospective Register of Systematic Reviews (PROSPERO, file number CRD42019119854).

### 2.1. Search Strategy and Study Selection

A librarian-assisted keyword search was used to identify relevant studies in the following databases: PubMed, Proquest Nursing and Allied Health, PsycINFO, SPORTDiscus and EMBASE via Ovid Healthstar. The following keyword search terms were used to guide study identification: “(((green gym[Title/Abstract] OR green exercise[Title/Abstract] OR outdoor physical activity[Title/Abstract] OR outdoor exercise[Title/Abstract] OR outdoor walking OR green physical activity[Title/Abstract]) AND (wellbeing OR WB OR mental health OR affect OR emotion OR self-esteem OR vitality OR purpose OR self-acceptance OR engagement OR personal growth OR satisfaction OR competence OR optimism OR resilience OR autonomy OR relationship OR eudaimonia OR flourishing)) NOT review AND (Humans[MeSH] AND English[lang] AND adult[MeSH]))”. Any database that restricted search options (e.g., no age limits, etc.) resulted in the use of a modified search strategy for this meta-analysis. DistillerSR [24] facilitated the study identification and selection process. Studies identified through the keyword search were first screened using title and abstract levels for inclusion by independent coders (N = 3). Studies meeting eligibility criteria at the title and abstract levels were retained before the full manuscript was retrieved then reviewed to determine eligibility of each study for inclusion in this meta-analysis consistent with PICOS (see Appendix A). Any study failing to meet inclusion criteria in this meta-analysis was omitted from consideration in subsequent analyses. Searches were completed in January 2019.

### 2.2. Data Extraction

A data extraction form was developed for use in this study based on the Cochrane Handbook 5.1.0 [25] and Consensus for Exercise Reporting Template (CERT) [26]. One point was awarded to each included study for every CERT criterion met. CERT scores were totalled, then converted into a value denoting the percentage of complete reporting. CERT items deemed irrelevant (e.g., home-based exercise, etc.) were coded as ‘not relevant’ and omitted from the total CERT score. Sample characteristics (e.g., gender, ethnicity, etc.) and study characteristics (e.g., design, instrumentation, etc.) were extracted from retained studies. Study quality was assessed with the Effective Public Health Practice Project (EHPP) Quality Assessment Tool for Quantitative Studies [27]. An overall quality score was generated for each study included in the data analysis. The first and second authors coded all primary studies retained for this meta-analysis. To maximize consistency, a subset of the primary studies (*n_studies_* = 5) were coded individually and any interpretational differences between coders discussed. Subsequently, both the first and second author proceeded to code all studies deemed eligible for this meta-analysis. Ongoing comparisons between coders were conducted on a week-to-week basis, amounting to ~5 primary studies per week for the duration of the coding process undertaken in this meta-analysis. Discrepancies were resolved by discussion between the first and second author. During the coding process for this investigation, no discrepancies between coders failed to attain resolution.

### 2.3. Data Analysis

Descriptive statistics for sample characteristics and study characteristics were calculated. All analyses were performed using Comprehensive Meta-Analysis (CMA) Version 3.0 [28] with a random effects model. Standardized mean differences (Cohen’s d) [29] were used to calculate effect sizes for included studies [30]. In studies that assessed more than one dimension of wellbeing (*n* = 9), a combined effect size was calculated in lieu of treating each dimension separately, given concerns reported in previous research of improper effect size estimates and relative weighting (i.e., overweighting certain studies) [30]. Calculations of effect sizes for pre–post-test research designs require correlations between the wellbeing dimensions measured at both time points. When provided in the primary source, or calculated, the correlations between wellbeing assessed pre–post-test were manually imputed to the CMA program. In lieu of correlations between pre- and post-test assessments of wellbeing, sensitivity analyses were conducted with different plausible values (specifically r_xy_ = 0.1 or r_xy_ = 0.5 or r_xy_ = 0.9) for the correlations between wellbeing at both time points [30]. Results are reported then interpreted with a fixed value for these correlation coefficients (i.e., r_xy_ = 0.5). Forest plots were created to provide a visual display of (a) the overall effect size across studies and (b) effect sizes per individual study. Heterogeneity was assessed for coded studies included in this meta-analysis using the following indices: (1) Cochran’s *Q* [31] as an assessment of true (i.e., between study) heterogeneity of variance; (2) *I^2^* to determine the percentage of total variance due to heterogeneity rather than chance across studies [32]; and (3) Kendall’s *T* and (τ^2^) or between studies variance [30]. Prediction intervals *(95% PI*) to assess the dispersion of true effect sizes as well as 95% confidence intervals (*95% CIs*) to quantify the precision of effect sizes were calculated [30].

## 3. Results

### 3.1. Search Strategy and Study Selection 

The electronic searches resulted in 1067 primary studies, with 19 studies [17,33,34,35,36,37,38,39,40,41,42,43,44,45,46,47,48,49] meeting all inclusion criteria after full-text inspection for the main aim of this meta-analysis (see Appendix A). Five studies [39,40,45,47,49] met all inclusion criteria for the secondary aim of this meta-analysis.

### 3.2. Main Aim: Effect of PA in the NOE on Wellbeing

*Sample and Study Characteristics*. The sample characteristics and study characteristics are presented in Appendix A. Sample sizes ranged from 6 to 263 participants (*M* = 42.05; *SD* = 56.16). The authors of 13 studies reported sample age (*M* = 31.95 years; *SD* = 12.30 years). A total of 11 studies reported the sample gender, with approximately two-thirds self-identifying as ‘female’ (*M* = 62.91%; *SD* = 19.97%). The authors of 5 studies reported participant ethnicity with 85.40% (*SD* = 14.96%) self-identifying as ‘White/Caucasian’. Publication dates of the primary studies ranged from 1995 to 2018. Most studies (*n* = 13; 68.42%) were coded as using quasi-experimental designs. NOE included one of four categories: (a) urban greenspace (*n* = 11; 57.89%), (b) mixed NOE (*n* = 5; 26.31%), (c) woodland/forest (*n* = 2; 10.52%), or (d) freshwater blue space (*n* = 1; 5.26%). The duration of PA ranged from 10 min to 230 min (*M* = 48.78 min; *SD* = 55.71 min) across coded studies with authors of one study not reporting values for duration of PA. Walking emerged as the most frequent mode of PA undertaken in the NOE reported in coded studies (*n* = 11; 57.89%). Multiple dimensions of wellbeing were used across coded studies, with vitality (*n* = 11; 57.89%) and positive affect (*n* = 10; 52.63%) the most frequently reported. All studies included in the analysis scored a ‘3′ indicating ‘low’ study quality based on criteria in the EHPP Quality Assessment Tool [27]. Intervention reporting based on CERT guidelines [26] varied considerably across coded studies, ranging from 38.40% to 93.30% (*M* = 71.80%, *SD* = 15.27%). 

*Effect Size Estimates*. A moderate overall effect size was observed (*d* = 0.49; *95% CI* = 0.33, 0.66, *p* < 0.001; *95% PI* = −0.17, 1.15; see Table 1 and Figure 1). Sensitivity analyses did not appreciably alter the magnitude or interpretation of conclusions. Regardless of the dimension of wellbeing assessed within coded studies, the effect size was statistically significant. Changes in ‘positive affect’ due to PA in the NOE demonstrated the largest effect size (*d* = 0.56; *95% CI* = 0.28, 0.84, *p* < 0.001; 95% *PI* = −0.42, 1.54), whereas ‘engagement’ displayed the smallest effect size (*d* = 0.30; *95% CI* = 0.02, 0.59, *p* = 0.03, *95% PI* = −0.77, 1.37). Effect size estimates by individual studies and dimensions of wellbeing can be found in Appendix A. Overall heterogeneity was significant (*Q* = 68.72, *p* < 0.001), and *I^2^* was considered moderate-to-high [32]. Significant heterogeneity such as that observed in this meta-analysis can justify further examination into sample characteristics and study characteristics that predict variability in wellbeing (i.e., moderator analyses). These analyses were not conducted in this meta-analysis due to (1) concerns regarding attenuation due to measurement error resulting (in part) from the small number of studies retained for coding, and (2) departure from the minimum number of studies recommended per moderator (i.e., *n_studies_* = 10/moderator) [25,30].

### 3.3. Secondary Aim: Differences in Wellbeing as a Function of Being Physically Active in the NOE versus Indoors

*Sample and Study Characteristics.* The sample characteristics and study characteristics used to address the secondary aim can be found in Appendix A. Sample sizes ranged from 15 to 42 participants (*M* = 27.60; *SD* =10.78; *M_age_* = 28.39 years, *SD_age_* = 6.98 years). Four studies reported participants’ sex (80%), with approximately two-thirds self-identifying as ‘female’ (*M* = 65.75%, *SD* = 29.15%). Two studies (40%) reported the ethnicity of the sample, with 76.50% of participants self-identifying as ‘Caucasian’. Publication dates ranged from 2016 to 2018. The most common environment type was urban greenspace (*n* = 3; 60%). The duration of PA ranged from 10 to 230 min. Cycling was the most common form of PA reported in this subset of coded studies (*n* = 2; 40%). Vitality (*n* = 4) followed by positive affect (*n* = 3) were the most frequently measured dimensions of wellbeing in this subset of coded studies. All studies included in the analysis were given a score of ‘3′ indicating ‘low’ quality based on criteria in the EHPP Quality Assessment Tool [27]. Intervention reporting based on CERT guidelines [26] was, on average, greater than 78% (*M* = 78.62%, *SD* = 10.26%).

*Effect Size Estimates*. The magnitude of the overall effect size comparing wellbeing resulting from PA in the NOE (*d* = 0.53, *p* < 0.001, *95% CI* = 0.28, 0.78) versus indoor contexts (*d* = 0.28, *p* = 0.02, *95% CI* = 0.04, 0.51) did not yield statistically significant differences (*p* = 0.15; see Table 2 and Figure 2). Between-group differences between PA in the NOE versus indoor contexts in effect sizes for vitality (*n* = 4; *p* = 0.48) and positive affect (*n* = 4; *p* = 0.51) were not statistically significant. Appendix A presents the effect size estimates by individual study and instrument used to assess wellbeing. Heterogeneity estimates were not statistically significant in the coded studies used to address the secondary aim of this meta-analysis. 

## 4. Discussion

Building on previous research [12,16,18,19], this meta-analytic study was conducted to investigate (a) the effects of PA in the NOE on wellbeing, and (b) differences in wellbeing resulting from PA in the NOE compared with indoor environments. Overall, the results of this meta-analysis generally align with conclusions drawn from previous research [12,16]. To summarize, the findings of this study make it apparent that (a) PA in the NOE is linked with higher wellbeing, and (b) PA in the NOE confers no unique benefits on wellbeing compared with PA performed indoors. Future considerations for research, as well as limitations of this investigation, serve as the basis for discussion with the intent of advancing the evidence base focused on PA in the NOE.

One key finding emerging from this meta-analysis is the moderate positive effect on wellbeing reported from performing PA in the NOE which did not vary across dimensions. This observation—that PA in the NOE is linked with higher wellbeing—is not wholly consistent with previous research. For example, [19] reported benefits for some dimensions of wellbeing (e.g., anxiety and fatigue), but not all dimensions (e.g., attention, tranquility) attributed to PA. It is possible that variations in criteria defining wellbeing across these PA studies explain this anomaly, given that previous research has used variables lacking content relevance with current definitions of wellbeing [1,2,20]. Based on this argument, it seems reasonable to contend that the results of this study highlight the importance of conceptual clarity required in future research to advance the study of wellbeing in the NOE where adults can (and do) engage in PA. 

A cautionary note emerged from the results addressing the main aim of this study for advocates of PA in the NOE that warrants commentary. In brief, the observed values for the *95% PI* implies that future research examining the effects of PA in the NOE on wellbeing may generate effect sizes ranging from small negative values to large positive values. Bornstein et al. [30] contend that prediction intervals provide a bandwidth of plausible values that can include the true effect size—which, in this study, is represented by the effect of PA in the NOE on wellbeing. It is speculative to comment on factors that may impact the *95% PI* attesting to the effect of PA in the NOE on wellbeing. However, given the observed heterogeneity in coded studies used in this meta-analysis, it would seem that characteristics of PA behaviour itself (e.g., frequency, intensity, duration, etc.) and/or characteristics of the study participants (e.g., sex, ethnicity, occupational status, etc.) and/or characteristics of the research design (e.g., number and timing of wellbeing assessments) may be considerations. In any event, given the *95% PI* generated in this meta-analysis, it seems prudent that conclusions focused on the effects of PA in the NOE on wellbeing be tempered with caution.

Building on previous work focused on PA and wellbeing, the secondary aim of this investigation was to evaluate differences in wellbeing attributed to PA in the NOE compared against PA completed indoors. Previous research focusing on this issue has yielded equivocal findings [16,18,19]. In this study, no statistical differences in wellbeing emerged as a function of PA in the NOE compared with indoor settings, which supports Lahart et al.’s [16] suggestion that effects of PA on wellbeing maybe context-free. Taken together, the findings of this meta-analysis paired with the conclusions advanced by Lahart et al. [16] could be interpreted as suggesting that PA results in greater wellbeing regardless of context, thereby offering individuals a choice of settings to engage in PA. Based on this study, it is not entirely clear ‘if’ or ‘how’ context (NOE versus indoors) may impact dimensions of wellbeing; however, it seems evident that additional mechanisms (e.g., person’s choice of context, etc.) may deserve closer scrutiny and further evaluation (see Araújo et al. [50] for an overview).

The centrality of wellbeing as an endpoint for public policy [5,6] combined with links between wellbeing and better health [3,4] provides ample grounds to advance recommendations for additional research. We offer two considerations for this line of inquiry that directly build on the results of this meta-analysis. First, it is recommended that future research investigating changes in wellbeing attributed to PA in the NOE align the measurement of wellbeing to modern conceptualisations (see Marsh et al. [2] for details). Past research testing the effects of PA in the NOE on wellbeing have relied on (a) measuring wellbeing with variables lacking content relevance to current definitions, and/or (b) measured wellbeing using a restricted assortment of variables that do not represent the full conceptual bandwidth and complexity of this psychological construct. This may lead to a distorted view of the effects (or lack thereof) of PA in the NOE on wellbeing in the literature. Future research can address this issue by examining the full spectrum of dimensions comprising wellbeing—particularly those aligned with the eudaimonic tradition [4]—to advance the evidence favoring (or negating) the role of PA in the NOE as a conduit to wellbeing.

Second, it is recommended that future research investigating wellbeing effects attributed to PA in the NOE pay closer attention to characteristics of PA behaviour, as well as specific features of the NOE in the research design. Regarding PA itself, researchers should strive to carefully document various characteristics of PA—including, at minimum, frequency, intensity, and duration—such that any effects of these issues on wellbeing can be considered in replication and extension studies. Sylvester et al. [51] illustrate the importance of carefully documenting features of PA in research using wellbeing as the outcome of interest. Using day reconstruction methods, wellbeing was not associated with the frequency or duration of PA, although it was linked with the effort spent doing PA [51]. Regarding the NOE itself, Bamberg et al. [52] contend that few studies have considered ‘how’ people interpret features of the NOE in research focused on wellbeing. Consider the following as an illustration: Would you expect cycling on a stationary bike overlooking a sports field [39] to effect wellbeing in the same way as running outdoors in a woodland/forest environment [49]? It is recommended that future research focused on wellbeing in the NOE and PA heed Bamberg et al.’s [52] insights by considering people’s interpretations of features hosted within the NOE that likely effect wellbeing.

This study has several limitations that warrant consideration coupled with future directions to advance the study of PA in the NOE relative to wellbeing. First, methodological issues evident in the primary studies coded in this meta-analysis limit study conclusions. These include, but are not limited to, issues such as: (a) small sample sizes, (b) over-reliance on acute bouts of PA, (c) restricted use of different PA modes beyond ‘walking’, (d) overuse of urban green space as the NOE, and (f) restricted variability around intensity of PA. Researchers are encouraged to thoughtfully design studies to minimize the risk of bias and report interventions consistent with existing standards for exercise [26]. Second, few studies have assessed the effects of PA in the NOEto include multiple assessment points for the dimensions of wellbeing. It is now well documented that the trajectory of affective responses to exercise as a stimulus decrease as PA becomes more intense, then increases upon the termination of PA [37,53]. Bowler et al. [19] recommended using multiple tests to better understand the influence of PA in the NOE on dimensions of wellbeing; however, few researchers have adopted this approach to their work (e.g., Niedermeier et al. [45]). Future studies could address this issue by assessing dimensions of wellbeing at multiple time points before and after the intervention using PA in the NOE as the stimulus. Finally, the findings may have limited generalizability as a function of the samples reported in the primary studies coded for this meta-analysis. Barriers to the access and use of the NOE as an inclusive space for PA have been reported [54] and may represent cultural or historical biases impacting the results of this study. Future research focused on PA in the NOE and wellbeing may wish to explore more traditional/Indigenous versus western Eurocentric understandings of human relationships with the environment [55].

## 5. Conclusions

In summary, the main aim of this study was to examine the effect of PA in the NOE on wellbeing. The secondary aim of this study was to evaluate differences in wellbeing between PA in the NOE compared with indoor environments. To address these aims, a meta-analysis was conducted using 19 (main aim) and 5 (secondary aim) primary studies that met all inclusion criteria for this investigation. Overall, it appears that PA in the NOE is linked with higher wellbeing given past studies in this area; however, future work should not assume that this association is guaranteed and there is no compelling evidence to support the superiority of PA in the NOE for enhanced wellbeing compared against PA performed indoors. Caveats in the evidence base examining PA in the NOE and wellbeing have been identified and serve as a fertile ground for future research in this area, which may inform public policy targeting wellbeing as an important endpoint.

## Figures and Tables

**Figure 1 sports-10-00103-f001:**
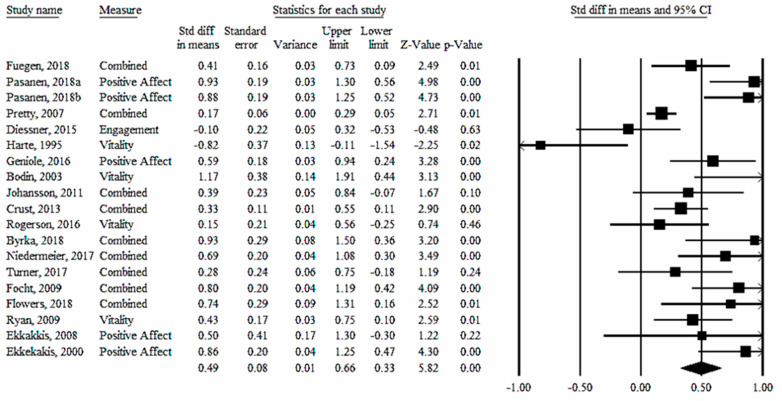
Forest plot for PA in the NOE on wellbeing.

**Figure 2 sports-10-00103-f002:**
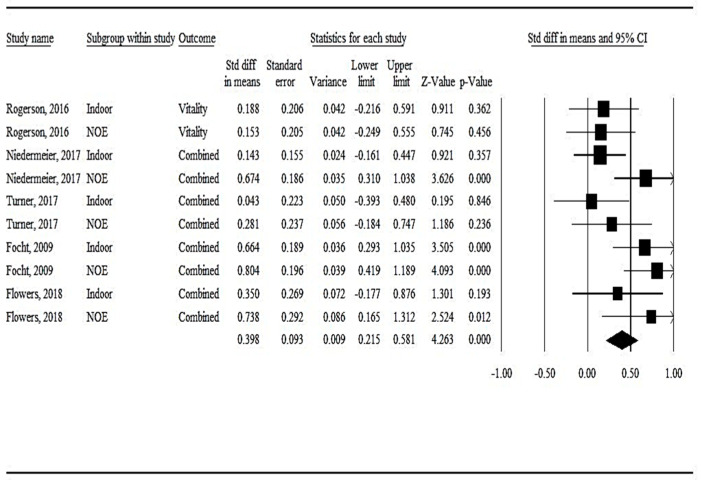
Forest plot for PA in the NOE compared with PA indoors on wellbeing.

**Table 1 sports-10-00103-t001:** Effect size estimates, heterogeneity and sensitivity analysis for the effect of PA in the NOE on wellbeing.

Measure	*k*	*ES*	*p-Value*	*95% CI* *[Lb, Ub]*	*95% PI* *[Lb, Ub]*	*Q*	*t* ^2^	*T*	*I* ^2^
Pos. Affect	10	0.56	0.00	[0.28, 0.84]	[−0.42, 1.54]	51.33 *	0.16	0.40	82.47
Vitality	11	0.52	0.00	[0.22, 0.82]	[−0.52, 1.56]	65.42 *	0.19	0.44	84.71
Engagement	4	0.30	0.03	[0.02, 0.59]	[−0.77, 1.37]	5.25	0.04	0.19	42.90
Self-Esteem	3	0.45	0.00	[0.22, 0.69]	[−1.93, 2.83]	5.27	0.02	0.16	62.05
Total (*r* = 0.5)	19	0.49	0.00	[0.33, 0.66]	[−0.17, 1.15]	68.72 *	0.09	0.30	73.81
Pos. Affect	10	0.53	0.00	[0.30, 0.75]	[−0.31, 1.37]	166.44 *	0.12	0.35	94.59
Vitality	11	0.48	0.00	[0.20, 0.76]	[−0.60, 1.56]	262.65 *	0.21	0.46	96.19
Engagement	4	0.27	0.07	[−0.02, 0.56]	[−1.03, 1.57]	20.83 *	0.07	0.27	85.60
Self Esteem	3	0.50	0.00	[0.29, 0.72]	[−3.15, 4.15]	22.19 *	0.03	0.174	90.99
Total (*r* = 0.1)	19	0.50	0.00	[0.33, 0.67]	[−0.09, 1.09]	42.13 *	0.07	0.26	57.28
Pos. Affect	10	0.57	0.00	[0.28, 0.85]	[−0.35, 1.49]	31.89 *	0.14	0.38	71.78
Vitality	11	0.53	0.00	[0.22, 0.83]	[−0.46, 1.52]	41.61 *	0.17	0.41	75.96
Engagement	4	0.27	0.02	[0.06, 0.61]	[−0.39, 1.07]	3.26	0.01	0.08	7.93
Self Esteem	3	0.41	0.00	[0.19, 0.63]	[−1.5, 2.32]	3.00	0.01	0.12	33.29
Total (*r* = 0.9)	19	0.44	0.00	[0.30, 0.59]	[−0.17, 1.06]	238.79 *	0.08	0.29	92.46

Note. *k*, number of studies; *ES*, standardized mean difference (Cohen’s *d*); *95% CI [Lb, Ub]*, lower and upper bounds of the 95% confidence interval; *95% PI [Lb, Ub]*, lower and upper bounds of the 95% prediction interval; *Q*, total variance heterogeneity statistic; *t**^2^*, variance of the true effect size; *T*, standard deviation of the true effect; *I^2^*, index of heterogeneity. Positive emotion was included in the combined wellbeing calculations, but omitted because only one study used this measure of wellbeing. * *p* < 0.05.

**Table 2 sports-10-00103-t002:** Effect size estimates, heterogeneity and sensitivity analysis for differences in wellbeing depending on PA in the NOE vs. indoors.

Measure	*K*	*ES*	*p-Value*	*95% CI* *[Lb, Ub]*	*95% PI* *[Lb, Ub]*	*Q*	*τ* ^2^	*T*	*I* ^2^
NOE	5	0.53	0.00	[0.28, 0.78]	[−0.22, 1.28]	7.52	0.04	0.20	46.83
Indoor	5	0.28	0.02	[0.04, 0.51]	[−0.30, 0.86]	6.28	0.02	0.15	36.29
Total Between (*r* = 0.5)						2.08			
NOE	5	0.55	0.00	[0.32, 0.79]	[0.05, 1.05]	4.52	0.01	0.10	11.53
Indoor	5	0.25	0.02	[0.03, 0.46]	[−0.09, 0.59]	3.80	0.00	0.00	0.00
Total Between (*r* = 0.1)						3.54			
NOE	5	0.53	0.00	[0.28, 0.78]	[−0.46, 1.52]	30.94 *	0.08	0.29	87.07
Indoor	5	0.29	0.02	[0.04, 0.54]	[−0.59, 1.17]	24.40 *	0.06	0.24	83.61
Total Between (*r* = 0.9)						1.74			

Note. *k*, number of studies; *ES*, standardized mean difference (Cohen’s *d*); *95% CI*
*[Lb, Ub]*, lower and upper bounds of the 95% confidence interval; *95%*
*PI [Lb, Ub]*, lower and upper bounds of the 95% prediction interval; *Q*, total variance heterogeneity statistic; *τ**^2^*, variance of the true effect size; *T*, standard deviation of the true effect; *I^2^*, index of heterogeneity; NOE, natural outdoor environment. * *p* < 0.05.

## Data Availability

Coding templates and data included for the review may be obtained by contacting [blinded for review].

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
