# Peer review of "Does Physical Activity in Natural Outdoor Environments Improve Wellbeing? A Meta-Analysis"

_sports, 2022, doi:10.3390/sports10070103_

Round 1

Reviewer 1 Report

Title

In the title, the authors ask whether PA is better for wellbeing. In the proposed form, the question seems unfinished because it is not clear what PA is better than. I suggest changing the title in "Does PA ... improve wellbeing?" of "Does PA ... have positive effect on wellbeing?"

Abstract

the d values shown in parentheses in one case have a zero before the decimal point, while in others they do not

Keywords

Please add "meta-analysis" as one of the keywords

Results

What is missing to understand and interpret your meta-analysis findings is that it is not clear how participants in the studies involved were selected: were they randomly invited, was their psychological health assessed prior to exercise, were they of average physical ability or really fit?

In 18 of the 19 studies, PA intensity was low/moderate/NR/self-selected/moderate to vigorous, while in the study by Harte & Eifert (1995), participants ran for 45 minutes and PA intensity was described as "vigorous". Is it possible that the participants in this research had too difficult task, so they assessed their wellbeing as worse than others? 

The result of this study and the result of the shortest, 10-minute study study by Diessner et al (2015), differ in wellbeing from other studies (Figure 1). Maybe ten minutes of walking was not long enough for participants to feel its benefits?

Heterogeneity in the duration of exercises in the included studies is also high and it is questionable whether a 10 to 15 minute moderate activity can result in an immediate sense of wellbeing in physically moderately active individuals.

If you still decide to keep these two studies in meta-analysis, please be sure to list their limitations.

The legend of Table 1 indicates a standard error, however this column is not shown in the table.

page 6, line 212 - there is a verb missing: "Cycling was the most common ..."

Reviewer 2 Report

The subject is very important nowadays, not only because of the low level of physical activity in general, but also because of the recent limitations that COVID has brought to us.

In my opinion, the work is very good, carefully done. From the introduction through the methodology to the discussion, it is difficult to find anything error.

Apart from minor mistakes, the work is ready for publishing.

L. 26 You do not need to use the phrase “physical activity” in here, it is enough to write “PE”

L. 56 change „{„ mark for „[„ mark.

What is the letter S used for before each successive table number?

CONGRATULATIONS.
